# A Model Selection Approach for Time Series Forecasting: Incorporating Google Trends Data in Australian Macro Indicators

**DOI:** 10.3390/e25081144

**Published:** 2023-07-30

**Authors:** Ali Abdul Karim, Eric Pardede, Scott Mann

**Affiliations:** Department of Computer Science and Information Technology, La Trobe University, Melbourne, VIC 3086, Australia

**Keywords:** convolutional neural network, feature selection, forecast comparison, Google Trends, Internet search data, SARIMAX, support vector regression, time series forecasting, tourism demand forecasting, unemployment

## Abstract

This study examined whether the behaviour of Internet search users obtained from Google Trends contributes to the forecasting of two Australian macroeconomic indicators: monthly unemployment rate and monthly number of short-term visitors. We assessed the performance of traditional time series linear regression (SARIMA) against a widely used machine learning technique (support vector regression) and a deep learning technique (convolutional neural network) in forecasting both indicators across different data settings. Our study focused on the out-of-sample forecasting performance of the SARIMA, SVR, and CNN models and forecasting the two Australian indicators. We adopted a multi-step approach to compare the performance of the models built over different forecasting horizons and assessed the impact of incorporating Google Trends data in the modelling process. Our approach supports a data-driven framework, which reduces the number of features prior to selecting the best-performing model. The experiments showed that incorporating Internet search data in the forecasting models improved the forecasting accuracy and that the results were dependent on the forecasting horizon, as well as the technique. To the best of our knowledge, this study is the first to assess the usefulness of Google search data in the context of these two economic variables. An extensive comparison of the performance of traditional and machine learning techniques on different data settings was conducted to enable the selection of an efficient model, including the forecasting technique, horizon, and modelling features.

## 1. Introduction

Forecasting the trends of economic indicators is crucial to policy makers and investors to make informed decisions. However, the official release of the indicators suffers from an information time lag because of the time and effort needed to collect the required data. To address this issue, researchers have aimed to nowcast and forecast the economic indicators.

The unemployment rate is one of the key indicators due to its direct connection to the economic cycle and its influence on decision-makers. Several researchers have attempted to improve the forecasting for the unemployment rate for various developed and developing countries. While some authors have applied different machine learning techniques to forecast unemployment [1,2], others have focused on incorporating additional data, in particular online search data, to improve the forecasting accuracy. Ettredge et al. [3] were the first to address such issues and investigated the link between online job searches and the official rates of unemployment in the United States. Additionally, Choi and Varian [4,5] put forward this line of research by describing and illustrating how Internet search data could be used to improve the predictions of several economic indicators such as unemployment claims, retail sales, property demand, and holiday destinations popularity. These two papers have stimulated much recent research in this field. Since none of the researchers investigated the relation between online search data and the unemployment rate in Australia, we chose to assess whether the search behaviour of Australian Internet users can improve the performance of the Australian monthly unemployment-rate-forecasting models.

In addition to the unemployment rate, we selected another indicator for our experiments, the number of short-term travellers visiting Australia. Being a destination for millions of tourists, the tourism industry in Australia is directly linked to its economic wellbeing. Forecasting the number of incoming travellers will assist investors in making their investment decisions and government agencies to properly allocate their resources to accommodate the number of travellers. Researchers have used online search data for different applications within the tourism industry. While some have focused on forecasting the hotel demand for particular cities or countries [6,7,8], others such as Feng et al. [9] and Gawlik et al. [10] have assessed the effectiveness of search data in forecasting the number of tourists rather than hotel demand.

The selection of the two indicators analysed in this study, which are released monthly by the Australian Bureau of Statistics, was based on their ability to reflect the behaviour of Google users across different geographical locations. While Google Trends data collected within Australia were used to forecast the monthly unemployment rate, we employed globally searched keywords via the Google engine to forecast the number of travellers visiting Australia. This approach enabled us to assess the applicability of Google Trends data for two distinct settings and evaluate the forecasting horizon associated with the behaviours of both local and international users. Furthermore, we present a novel forecasting framework that selects the optimally performing model from two families of techniques suitable for forecasting time series data, namely traditional linear techniques (SARIMA and SARIMAX) and machine learning techniques (SVR). The framework also incorporates feature selection techniques, which play a crucial role in the forecasting process. It should be noted that prior literature had not extensively explored this aspect to the extent that is presented in this paper.

In our paper, we examined the predictive power of Google Trends data using support vector regression (SVR) and convolutional neural networks (CNNs) against the traditional linear regression techniques such as SARIMA and SARIMAX in forecasting the two selected time series indicators. The paper is organised as follows. Section 2 presents an overview of the literature on using Google Trends data for economic indicators. Section 3 presents our contribution of applying a data-driven approach to forecast both indicators through a description of the methodology covering the data collection, feature engineering and selection techniques, as well as the forecasting models used in our paper, alongside the evaluation metrics. Section 4 describes the experimental setup, in particular a description of each set of experiments and the datasets associated with them. Section 5 evaluates empirically the forecasting performance of our models and provides an indication as to why a data-driven approach to forecasting is necessary. Section 6 contains a discussion on the suitability of using alternative data and non-traditional techniques for forecasting.

## 2. Literature

Over the last few years, several attempts have been made to explore the potential benefits of using Internet search data in forecasting economic variables [4]. In this section, we present some of the recent studies that have incorporated Internet search data in unemployment and tourism demand forecasting. To the best of our knowledge, our paper is the first to assess the usefulness of Google search data in the context of these economic variables in Australia and the first to compare the performance of traditional and machine learning techniques on different data settings.

### 2.1. Unemployment Forecasting

Forecasting unemployment has become an area of interest for researchers. There are two areas of focus to improve its accuracy: incorporating additional data sources (mainly Internet search data) and using non-traditional techniques.

Incorporating online search data, in particular Google Trends, in unemployment forecasting has drawn researchers’ attention. Evidence can be found for research applied in different countries: the U.S. [4,11,12,13], the U.K. [14], Italy [15,16], Canada [17], Brazil [18], Spain [19,20], France [21], Germany [22], Israel [23], Norway [24], Romania [25], China [26], Turkey [27], Ukraine [28], Poland and Slovakia [29]. Those papers improved the performance of their forecasting models by employing keywords related to employment such as “job”, “career”, “CV”, “job offer”, and “unemployment insurance” across different demographic settings. Alternatively, Dilmaghani [17] found that using keywords related to some online leisure activities explained the unemployment rate in Canada for people between 25 and 44 years old better than the keywords previously mentioned.

These studies did not establish whether Internet data can replace or complement traditional methods. Some authors obtained better results when combining both data in their model [16]. Most researchers have suggested using multiple keywords to improve the prediction accuracy of the forecasting models. In this paper, we compared the models using combined data, as well as search data on their own to address this limitation.

There is another set of research focused on using alternative techniques to forecast unemployment. Researchers have compared several machine learning techniques such as artificial neural networks (ANNs) [2,30,31], SVR [2], as well as hybrid approaches [1]. They found that their experiments yielded better results than ARIMA models.

Considering that researchers have not tested the impact of search data when forecasting unemployment using traditional and machine learning techniques, we were interested in assessing the efficacy of Google Trends in Australia where Google search is widely used. For this forecasting purpose, we employed SARIMA, SVR, and CNNs on an expanded list of search keywords that is related to Australia.

### 2.2. Tourism Forecasting

The real-time characteristics of Internet search data have motivated researchers to examine their predictive power in the tourism and hospitality industry. The scope of past research varied from forecasting hotel demand to the number of visitors to cities and countries.

Several research works have successfully employed Internet data to forecast the demand for hotel rooms and flights for different forecasting horizons [6,7,8,32]. Similar to unemployment forecasting, tourism research has been extended to predict the volume of visitors to cities [33,34] and countries [5,9,10,25,26]. Their results presented higher accuracy when incorporating search data. A limited number of those research works focused on the volume of incoming visitors regardless of their point of departure and did not evaluate the performance of other techniques.

While fewer studies have forecasted the number of visitors on a macro level, there has not been an assessment of the benefit of using search data with historical visitors’ data using machine learning techniques. In our paper, we used the same approach applied on unemployment data to evaluate the SARIMA and SVR results in forecasting the number of short-term visitors coming to Australia. A similar comparison was performed recently by Botta et al. [35], but instead of using SVR, they deployed an ANN to predict the number of a local museum visitors. We also applied the search keywords used by Feng et al. [9] and tailored them to the Australian context, since they were proven successful in forecasting the number of visitors and they covered different aspects of tourism (food, airline, shopping).

## 3. Methodology

### 3.1. Data Collection

The initial stage of our research involved data collection. We utilised two main sources of data: Australian economic indicators and Google Trends data. For the economic indicators, we extracted the historical data of two key indicators for the Australian economy from the Australian Bureau of Statistics’ website: monthly unemployment rate and monthly number of short-term visitors arriving in the country. These indicators represent essential aspects of the economy, and forecasting their future values would offer significant insights for policymakers and economic stakeholders. Those figures are often calculated by conducting surveys and collecting data from different agencies, leading to a delay in publishing the most-recent numbers. Monthly unemployment rate data are available from February 1978, while the number of visitors’ data cover the period starting in January 1991.

Australia has a stable economy. The unemployment rate has not surpassed the 10% mark since 1994. Since then, the Australian unemployment rate has fluctuated between 4% and 6%. As seen in Figure 1, there were two spikes/increases in unemployment in the last two decades: once after the GFC and one during COVID-19.

Australian unemployment data are seasonal in nature, where the same trend is repeated each year. For example, there has always been an increase in the unemployment rate post December, and this is expected to continue in the future. Since we used the SARIMA model, the parameter m that indicates the cycle of the trend was 12.

Figure 2 shows the number of short-term visitors coming to Australia. Australia is becoming a more-popular destination over time. The seasonality in the data is visible through the repeated trends.

A closer inspection of Figure 2 shows that the same trends are repeated the same month every year, e.g., an increased number of visitors around Christmas time and during summer. The large drop of the number of visitors on the right-hand side of the chart is due to COVID, when Australia had travel restrictions in place.

In parallel, we collected Google Trends data related to the aforementioned economic indicators. Google Trends data, which have been offered by Google since 2014, provide the search frequency of keywords, which shows the ratio of the search amount of a certain keyword to the total search amount of all keywords in a certain period of time, and then further normalises the search frequency into the interval of [0, 100], which can avoid changes in the amount of keyword searches due to an increase in the number of users. It represents a rich source of insights about public interest in various topics over time. By selecting search terms related to the economic indicators, we could gauge public interest in these topics and examine the potential predictive power this interest holds for future economic conditions.

In this paper, we searched for keywords related to each of the two target indicators used in this paper. For unemployment, the process of selecting search keywords began by considering what Internet users would search for if they became or were about to become unemployed [4]. It seems sensible to suggest that our searches were likely to be focused on two areas: available benefits to the unemployed and particular websites and keywords that unemployed people may use (e.g., job advertisement website, “job and education” topic search).

Table 1 shows the data extracted from the Google Trends service. Centerlink is an Australian government service that offers several benefits including unemployment benefits. Additionally, we incorporated an indicator for “job” to accommodate searches related to job searches that are general in nature and difficult to capture using more specific terms. Seek and Indeed are popular job advertisement websites, which are mainly used to look for job vacancies, so they are also included. Additionally, we added the trend data of searches for the word “unemployment”. All the extracted data using Google Trends were restricted to searches within Australia. Figure 3 shows the popularity of four of the Google indicators extracted to forecast the unemployment rate.

There were some limitations associated with selecting search keywords relevant to unemployment. Centerlink offers several services other than unemployment benefits; therefore, a change in its trend does not necessarily reflect the changes in demand for those benefits. Additionally, there are certain job vacancies relevant to industries such as construction that might not be posted on the “Seek” website. An increase in unemployment in the construction industry might not lead to an increase in access to the popular job search website. Furthermore, there are other popular platforms such as LinkedIn that can be accessed via a mobile application or directly through the website to look for job vacancies.

Given the limitation of using Google Trends data, we intended to use the extracted time series data as a proxy for changes in the labour market, rather than an accurate reflection of changes in the Australian unemployment rate.

The selected Google indicators to be used in forecasting the number of short-term visitors is shown in Table 2 alongside their reference names used in our code. Those indicators are similar to those used by Feng et al. [9], and they cover different areas of what travellers might need to get to their destination and to facilitate their visit. Terms such as “Australian weather” and “Australian climate” indicate the interest of search engine users in knowing what to wear when visiting Australia. The terms “Australia airline”, “Qantas”, and “Australian map” indicate the interest to know more on how to get to and navigate Australia; Qantas is the flagship carrier of Australia and its largest airline by fleet size, international flights, and international destinations. The extracted search data cover worldwide searches in contrast to the ones used to forecast unemployment, which were restricted to Australia.

One of the limitations of using keywords looked up all over the world is that this includes the searches of users within Australia. Searches for those keywords by Australian residents do not contribute to the number of tourists visiting Australia. For this exercise, we assumed that the search for these terms within Australia did not create any noise as there were no noticeable changes in the search trends. Additionally, the search results were limited to the Google engine and did not include the usage of people residing in China due to the restriction on using Google in China. Chinese nationals consist of large proportion of tourists visiting Australia.

### 3.2. Feature Engineering

After the data collection, we proceeded to the feature-engineering phase. The goal was to transform the collected data into a format that could be more effectively utilised by our predictive models. This involved creating new variables based on our raw data that better represent the underlying trend patterns for the predictive models. This process was applied in our study to increase the predictive performance of our models.

For our dataset from the Australian Bureau of Statistics (ABS) and Google Trends, the original data were augmented by creating time-based features. These features were designed to capture the dynamic behaviour and trends in the data over time. These included lagged values of the indicators themselves and derived statistics such as moving averages.

We created lag features for each dataset, specifically for the 12 previous months. The assumption here was that the current month’s value of a given economic indicator (such as the unemployment rate or visitor arrivals) or Google Trends value would have some correlation with its past values. For instance, if the unemployment rate was high last month, it could likely be high in the current month as well, barring any substantial changes in the economic environment.

Lagged features were derived by shifting the time series data by one period (month) to create a new feature (Lag-1), by two periods to create another feature (Lag-2), and so on, up to twelve periods (Lag-12). This was carried out because it is plausible that both the dependent variables and the Google Trends indicators could have monthly seasonality that last up to a year, and we wanted our models to capture this potential seasonal effect.

We also created moving average features, which represent the mean of the data points over a specified period. These were calculated for the last 3, 4,…, and 12 months. The rationale for creating these features is that, while individual data points (such as a spike in search interest or a dip in unemployment) can be quite volatile, the average value over a certain period can provide a smoother representation of the underlying trend in the data.
$$X_{average\left(n\right)}=(X_{lag\left(i\right)}+X_{lag\left(i+1\right)}+\ldots +X_{lag\left(n\right)})/n$$

In addition to the lag and moving average features, a “month” feature was created to capture any potential seasonal effects. This feature represents the month of the year (a number between 1 and 12) at each data point. This is particularly important for data such as tourism, which can show substantial variation depending on the time of year. Table 3 shows a list of all the features created.

The high-volatility components associated with time series data are often very difficult to model successfully; hence, a scaling and/or transformation process is usually performed on the series prior to implementing the actual experiments [36].

Since we wished to be able to correctly predict the direction of movement of the number of short-term visitors, we applied a data transformation to the data series, which would result in better performance [37]. Natural logarithm transformations were applied to the data series prior to conducting the SARIMA(X) and SVR algorithms.

To achieve a logarithmic transformation with our short-term visitors’ data, the following equation was applied.
$$y_{t}=\ln\left(p_{t}\right)$$
where *y_t_* is the transformed number of visitors and *p_t_* is the original value.

### 3.3. Feature Selection

In recent years, many feature-selection methods have been proposed. These methods can be categorised into three [38]: filter, wrapper, and embedded methods.

Filter methods calculate the score of each feature and rank them accordingly without dependency on the model. They are simple to implement, easy to interpret, and work effectively with high-dimensional data. Filter methods are fast strategies that provide good results in classification tasks [39,40,41]. An extensive overview of existing filter methods was presented by Lazar et al. [42].

After engineering a wide range of features from the target variables and Google Trends indicators, we applied different feature-selection methods that incorporated recursive feature elimination (RFE) with mutual information (MI) and the f_test. These methods provided us with a robust and diverse perspective on feature importance. For the exogenous variables derived from Google Trends, we used the Pearson correlation to determine the most-relevant variables, which were used to train the SARIMAX model.

The wrapper method, RFE, uses a machine learning algorithm (in our case, a DecisionTreeRegressor) to rank features by importance and recursively eliminates the least-important features. This method can capture interactions between features since it uses a machine learning model for ranking.

The filter methods, the f_test and mutual information, rank features based on their individual predictive power. The f_test checks the correlation between each feature and the target variable, while mutual information measures the dependency between the feature and the target. A higher mutual information means a higher dependency.

By using these methods together, we obtained the benefits of both: the power of a machine learning model to capture complex relationships and the speed and simplicity of univariate statistics.

The filter feature selection approach used for the SVR and CNN models is shown in Figure 4 and described in the snippet below.


*Create a training dataset.*

*Perform RFE using a decision tree as an estimator.*

*Select the top 50% of the features from RFE.*

*Compute the mutual information value (MIV) and f_test for the remaining features.*

*Filter out the features based on the f_test and MIV. Select the top 10% of features based on the f_test and the top 25% based on the MIV.*


The approach of using the Pearson correlation as a feature-selection method for our SARIMAX model is a straightforward, yet effective one given that SARIMAX is not capable of modelling non-linear relationships.

The Pearson correlation coefficient measures the linear relationship between two datasets. It ranges from −1 to 1. A correlation of −1 indicates a perfect negative linear relationship; a correlation of 1 indicates a perfect positive linear relationship; a correlation of 0 indicates no linear relationship.

In our experiment, we selected only those exogenous variables that have a correlation value greater than 0.4 (either positive or negative) and considered to have a moderate to strong linear relationship with the dependent variable. This could help reduce the dimensionality of our data and might improve the interpretability and performance of our models.

In summary, we chose a combination of feature selection and reduction techniques in our experiments to highlight the importance of incorporating such techniques in the modelling process to improve the accuracy of the models. The comparison of different techniques is out-of-scope for this paper. However, the selected techniques can detect different relationship between the created features and the target variable.

### 3.4. Forecasting Techniques

The seasonal auto-regressive integrated moving average (SARIMA) is an extension of the ARIMA model. ARIMA models are a subset of linear regression models that attempt to use the past observations of the target variable to forecast future values. The “S” in SARIMA stands for seasonal. It adjusts the model to deal with repeated trends. Seasonal data can be easily identified by looking at repetitive spikes over the same period of time. Those spikes are consistently cyclical and easily predictable, which suggests that we should look past the cyclicality to adjust for it.

Since SARIMA can only use the past values of Y and X, SARIMAX is used to incorporate exogenous variables. When using SARIMAX, the input data will include parallel time series variables that are used as a weighted input to the model.

To find the optimal SARIMA and SARIMAX models, a grid search to determine the value of the parameters for the best model was performed. The best model found will have the lowest Akaike’s information criterion (AIC) and Bayesian information criterion (BIC).

SARIMA and SARIMAX were used as the baseline models for the time series forecasting of the two Australian indicators of interest: monthly unemployment rate and monthly number of short-term visitors.

Since the SARIMAX model can only detect the linearity between the target variable and the past values of the input data, we employed SVR and CNNs to check whether there was non-linearity between the input feature and the target variable, and therefore, the forecasting performance can be improved over that of SARIMAX.

SVR, introduced by Drucker et al. [43], is a category of the support vector machines (SVMs), originally introduced by Vapink [44]. The model produced by SVR only depends on a subset of the training data, because the cost function for building the model ignores any training data that are close (within a threshold ε) to the model prediction. A detailed analysis and description of SVR can be found in Basak et al. [45], Sapankevych and Sankar [46], and Smola and Schölkopf [47] and an application to the prediction of unemployment in Stasinakis et al. [48]. SVR has been used widely for time series prediction [46], and the application areas are many, such as financial forecasting [49], among others.

Convolutional neural networks (CNNs) were introduced by Yann LeCun, Yoshua Bengio, and others in the 1990s [50]. Initially, CNNs were primarily developed and used for computer vision tasks such as image classification. However, CNNs have also been adapted and applied to other domains, including time series analysis and regression tasks. While CNNs were originally designed for image-based data, their ability to learn hierarchical patterns and capture local dependencies in data makes them suitable for analysing time series data as well. In time series analysis, CNNs can be used as regression techniques by applying them to the input data and predicting the target variable. By leveraging the convolutional layers and pooling operations, CNNs can automatically learn and extract relevant features from the time series data, making them powerful tools for time series forecasting and regression tasks.

To carry out non-linear regression using SVR and CNN, it is necessary to create a higher-dimensional feature space from the time series data, as discussed in Section 3.2.

### 3.5. Model Evaluation

In order to evaluate the performance of the SARIMA, SVR, and CNN models on out-of-time sample data, we used two different metrics: mean-squared error (MSE) and symmetric mean absolute percentage error (SMAPE) [51]. These two metrics proxy the accuracy of the model since they distinctly measure the difference between the actual and predicted values. The objective of our experiments was to improve the accuracy of the models; therefore, they seemed appropriate to evaluate the results.

The *MSE* is a metric corresponding to the expected value of the squared error or loss. If *ŷ_i_* is the predicted value of the i-th sample and *y_i_* is the corresponding true value, then the *MSE* estimated over *n* (number of samples) is defined as:(1)$$MSE\left(y,\hat{y}\right)=\frac{1}{n_{samples}}\overset{n_{samples}-1}{\underset{i=0}{\sum }}{\left(y_{i}-\hat{y_{i}}\right)}^{2}$$

The *SMAPE* is an accuracy measure based on percentage (or relative) errors, defined as follows:(2)$$SMAPE=\frac{100\%}{n}\overset{n}{\underset{t=1}{\sum }}\frac{\left|F_{t}-A_{t}\right|}{\left(|F_{t}|+\left|A_{t}\right|\right)/2}$$
where *At* is the actual value and *Ft* is the forecast value.

The absolute difference between *At* and *Ft* is divided by half the sum of the absolute values of the actual value *At* and the forecast value *Ft*. The value of this calculation is summed for every fit point t and divided again by the number of fit points *n*. A perfect *SMAPE* score is 0.0, and a higher score indicates a higher error rate.

Further statistical significance testing was applied to evaluate the performance of the different techniques and to determine if there were significant differences among them. One approach is to use the analysis of variance (ANOVA) on the predicted values generated by multiple models (ARIMA, ARIMAX, SVR, and CNN). ANOVA assesses the variation between the predicted values of different models and compares it to the overall variation in the data. The goal was to determine if there are statistically significant differences in the performance of the models.

After performing ANOVA, if significant differences are detected, further analysis can be conducted using post hoc tests to identify specific pairs of models that significantly differ from each other. One commonly used post hoc test is Tukey’s honestly significant difference (HSD) test. The Tukey HSD compares all possible pairs of models and determines if the differences in their predicted values are statistically significant.

The statistical significance approach helps with comparing and ranking the models based on their performance and identifying the models that significantly outperform or underperform others. It provides a quantitative and objective measure to assess the statistical differences between the techniques, allowing for informed decision-making in selecting the most-appropriate model for time series forecasting tasks.

## 4. Experimental Setup

In this paper, we sought to examine the out-of-sample forecast performance of the SARIMA, SVR, and CNN models with a focus on two key Australian indicators: unemployment rate and monthly number of short-term visitors. The methodology, delineated in Section 3, was consistently applied across all our experimental setups. Each setup entailed two distinct data periods, one considering all available data up to December 2022, and another ending in December 2019. This approach allowed us to make an equitable comparison between the models built using the full dataset versus those developed using a reduced data subset.

The design of our experiments was intended to assess the influence of the COVID-19 pandemic on the correlation between our chosen indicators and Google Trends data. By intentionally omitting data from the last three years and focusing on the pre-pandemic period, we evaluated if the dynamics between the indicators and Google Trends were dissimilar during a relatively more economically stable period.

In each experimental setup, we constructed four iterations of each of our 12 models (elaborated further in Table 4, Table 5 and Table 6). Each iteration was trained and tested on a different dataset corresponding to its unique forecasting horizon. We built two SARIMA models, one that utilised all the historical data and another that incorporated the data from 2005 onwards. The objective here was to assess whether the inclusion of more historical data enhanced the model’s performance. Subsequently, two SARIMAX models were developed, one utilising all exogenous variables and another using a subset of selected variables, as outlined in Section 3. This exercise allowed us to juxtapose the performance of SARIMAX with the SARIMA model constructed using more-recent data, as well as to discern if the Google Trends data could bolster the model’s accuracy. The SARIMAX model with selected exogenous features served as a comparison point with the original SARIMAX model.

Furthermore, we constructed the SVR and CNN models using all features from the target variable and then a subset of these features after implementing the RFE, MI, and f_test. This approach enabled us to contrast the performance of SVR and the CNNs with SARIMA and determine whether the feature selection enhanced the model performance. Later, the SVR and CNN models were constructed using all Google Trends features along with the target variable features for comparison with SARIMAX. The same models were then built using a selected subset of features. This exhaustive comparative analysis enabled us to assess the effectiveness of the machine learning and deep learning models vis-à-vis the conventional ones. It also helped ascertain if the incorporation of Google Trends data enhanced the predictive accuracy of these models and whether contemporary models more effectively encapsulated the relationship between the variables. Moreover, the utility of feature selection in improving outcomes could be gauged. Comparing different experimental sets provided insights into the influence of the Google data on the model performance, in particular by contrasting the model outcomes using datasets that include and exclude the COVID-19 period. 

## 5. Results and Discussion

In this section, we present an overview of the experiments’ results for each individual set of experiments. Additional comparison between Experiments 1 and 2 and Experiments 3 and 4 were conducted to highlight the difference in the performance between the models and the features selected for different data-driven settings influenced by COVID19. Given the large number of experiments and comprehensive statistical significance tests for the built models, only the comparison of results using the MSE are presented in Table 7, accompanied by the feature-selection results for each set of experiments in Table 8. The full results along with the data and code used to conduct these experiments are available at the following Git repository: https://github.com/a-abdulkarim/time-series-forecasting-p1/ (accessed on 1 July 2023).


*Experiment 1:*


The first experiment revealed significant differences in performance among the models across the four forecasting horizons. The SARIMAX_ALL model outperformed all others for the 3-, 6-, and 12-month horizon levels, indicating its strong predictive power in the short- to mid-term. Interestingly, the SARIMA_HIST model, utilising historical data without the inclusion of exogenous variables, performed better for the 24-month horizon, hinting at its efficacy in capturing long-term trends and cycles.

Compared to the SARIMA_RECENT, which takes into account only recent data, SARIMA_HIST’s superior performance for the 24-month horizon suggested that a broader historical context enhances long-term forecasting. SARIMAX_ALL’s outperformance of SARIMA_HIST and SARIMA_RECENT for shorter horizons demonstrated the value of integrating all available features, including exogenous variables, into time series models for short-term forecasts.


*Experiment 2:*


In the second experiment, the superiority of the SARIMAX_ALL model continued for the 6- and 12-month horizons, but faced competition from the CNN_TARGET_GI_FS model for the 3-month horizon. This indicates that deep learning models like CNN_TARGET_GI_FS can capture intricate data patterns more effectively in the short-term. For the 24-month horizon, however, the SARIMA_HIST model again outperformed, reaffirming the notion that simpler models utilising a broader historical context fare better in long-term forecasting.

Feature selection models, such as SARIMAX_FS and CNN_TARGET_GI_FS, performed comparably to their all-feature counterparts for shorter horizons, suggesting that narrowing down the feature set does not necessarily impair short-term predictive capacity.


*Experiment 3:*


The third experiment introduced a new dominant model: SVR_TARGET_GI_FS. This machine learning model with feature selection demonstrated the best performance at the 3- and 6-month horizon levels, outperforming both SARIMA variants and SARIMAX_ALL. This suggested that machine learning techniques coupled with feature selection can excel in short-term forecasts. However, the SARIMAX_ALL model still held its ground for the 12-month horizon, and SARIMA_HIST regained superiority for the 24-month horizon.

Again, feature selection models showed strong performance. The SVR_TARGET_GI_FS model’s superiority for shorter horizons over SARIMAX_ALL indicated that feature selection can even outperform all-feature models in certain situations.


*Experiment 4:*


In the final experiment, the deep learning model CNN_TARGET_GI_FS excelled for the 3- and 6-month horizons, while SARIMAX_ALL performed best for the 12-month horizon. For the 24-month horizon, the SVR_TARGET_FS model, a machine learning model with feature selection, surpassed other models, affirming the potency of feature selection for longer-term forecasting.

Across all four experiments, the results demonstrated the strengths and weaknesses of each model for different forecasting horizons, the potential advantages of machine learning and deep learning techniques over traditional SARIMA/SARIMAX models, and the possible gains from employing feature selection.

Taken together, these experiments provided nuanced insights into the interplay between traditional models (SARIMA and SARIMAX) and more modern, ML and DL techniques. While the former maintained strong performance at medium-term horizons, in particular when supplemented with a complete feature set, the latter—especially when utilising feature selection—appeared more-effective for both short- and long-term forecasting. Thus, the decision between ML/DL and traditional methods hinges on the forecasting horizon, underlining the importance of a targeted approach in time series prediction.

Compared to SARIMA_RECENT, which takes into account only recent data, SARIMA_HIST’s superior performance for the 24-month horizon suggests that a broader historical context enhances long-term forecasting. SARIMAX_ALL’s outperformance of SARIMA_HIST and SARIMA_RECENT at shorter horizons demonstrated the value of integrating all available features, including exogenous variables, into time series models for short-term forecasts.

### 5.1. Impact of COVID Event on Models Forecasting

#### 5.1.1. Unemployment Forecasting

Comparing both experiments, it was clear that the inclusion or exclusion of the COVID period data significantly influenced the predictive power of the models. In the shorter-term forecasts (3-, 6-, and 12-month horizons), the exclusion of the COVID period seemed to enhance the performance of models such as the CNNs, possibly due to the reduction of unprecedented volatility in the training data.

In contrast, the SARIMAX model, which was the most-effective short- and mid-term forecasting model when the COVID data were included, saw its dominance reduced when the COVID data were excluded. This indicated that the SARIMAX model might be particularly effective at accounting for abrupt exogenous shocks such as the COVID pandemic.

For the 24-month horizon, the SARIMA_HIST model remained the superior performer, with or without the COVID data, indicating its robustness in long-term forecasting regardless of drastic economic changes.

These comparisons highlight the importance of considering the stability of the economic environment and the characteristics of the training data when selecting and interpreting forecasting models.

#### 5.1.2. Number of Visitors Forecasting

Comparing both experiments, it became evident that the inclusion or exclusion of the COVID period data significantly impacted the models’ predictive performance. In the short-term forecasts (3-, 6-, and 12-month horizons), the exclusion of the COVID period data seemed to improve the performance of the CNN with the feature selection model, possibly due to the removal of the unpredictable COVID-induced volatility from the training data.

Conversely, the SARIMAX using the all exogenous variables model, which was the most-effective short- and mid-term forecasting model with the COVID data included, saw a reduction in its dominance when the COVID data were excluded. This indicated that the model was particularly potent when dealing with abrupt exogenous shocks such as those experienced during the COVID-19 pandemic.

For the long-term 24-month horizon forecast, the SARIMA_HIST model remained the best performer, irrespective of whether the COVID data were included or excluded, highlighting its robustness in long-term forecasting regardless of drastic changes in the economic environment.

These findings underlined the importance of considering both the stability of the economic environment and the nature of the training data when choosing and interpreting forecasting models. They also demonstrated how different models may respond differently to periods of economic volatility, further emphasising the need for careful model selection based on the specific context and forecasting horizon.

## 6. Conclusions

This research investigated the efficacy of various traditional and machine learning models in forecasting key economic indicators, namely the monthly unemployment rate and the monthly number of short-term visitors to Australia. It also explored the role of Google Trends data in enhancing the forecasting performance of these models.

Overall, the results indicated that both machine learning (ML) and deep learning (DL) models offer considerable advantages over traditional SARIMA and SARIMAX models in forecasting these indicators, particularly in the shorter-term forecasting horizons. For instance, the SVR model demonstrated superior performance over SARIMA and SARIMAX in predicting the unemployment rate across all forecasting horizons in Experiments 1 and 2. Similarly, the CNN model was more effective than its traditional counterparts in predicting short-term visitor numbers in Experiments 3 and 4, especially in the short- to mid-term forecasting horizons.

These findings align with the growing recognition of ML and DL techniques as valuable tools in economic forecasting, capable of handling complex data structures and identifying intricate patterns in the data. However, the results also underscored the robustness of traditional models such as SARIMA and SARIMAX in long-term forecasting, reminding us of their enduring relevance in certain forecasting contexts.

Importantly, the inclusion of Google Trends data proved to enhance the forecasting performance of several models. Models incorporating Google Trends data, such as SARIMAX and the CNN with feature selection, consistently outperformed their counterparts that relied solely on historical data, particularly in the short- to mid-term forecasting horizons. These findings affirmed the potential of Google Trends data as a valuable supplement to traditional economic data, particularly in an era where digital information plays an increasingly central role in economic activities.

This study, however, was not without its limitations. The forecasting performance of the models might be sensitive to the inclusion or exclusion of extreme events, such as the COVID-19 pandemic period data. The volatility introduced by such events can impact the predictive capability of different models in various ways, making it difficult to ascertain the most-effective model across all possible contexts.

Furthermore, while the study considered a broad range of models and data types, there are still other potentially useful models and data sources that remain unexplored. For instance, other types of ML and DL models, such as recurrent neural networks (RNNs) and transformers, might offer different insights or outperform the models investigated in this study.

Future research should aim to address these limitations and explore these uncharted territories. More-comprehensive investigations could consider a broader range of extreme events and their impacts on different models or investigate other types of ML and DL models and their efficacy in forecasting economic indicators. Moreover, future studies could explore other types of auxiliary data, such as social media data or other online data, to gauge their potential in enhancing economic forecasts.

In conclusion, this research underscored the potential of ML and DL techniques in economic forecasting and highlighted the value of integrating Google Trends data into these models. However, it also stressed the importance of model selection based on the specific forecasting context and the need for the continuous exploration of novel models and data sources to enhance our forecasting capabilities.

## Figures and Tables

**Figure 1 entropy-25-01144-f001:**
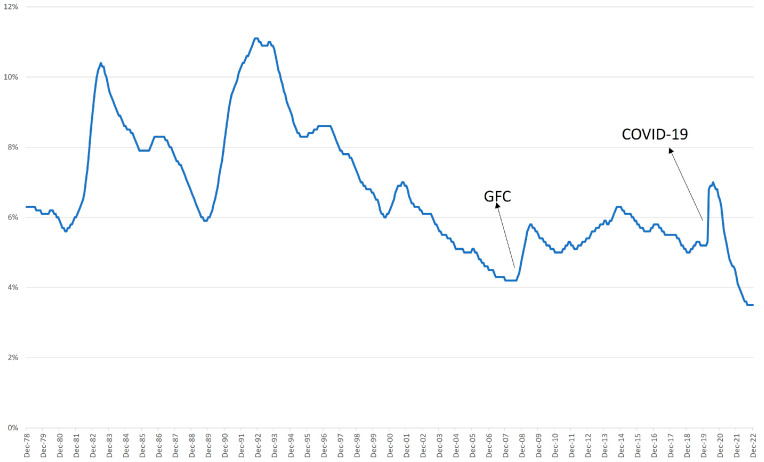
Australian unemployment rate (monthly percentage).

**Figure 2 entropy-25-01144-f002:**
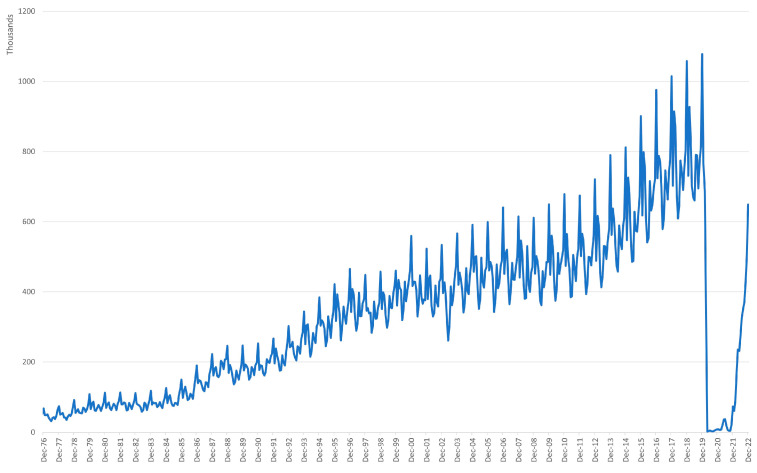
Number of short-term visitors entering Australia (monthly).

**Figure 3 entropy-25-01144-f003:**
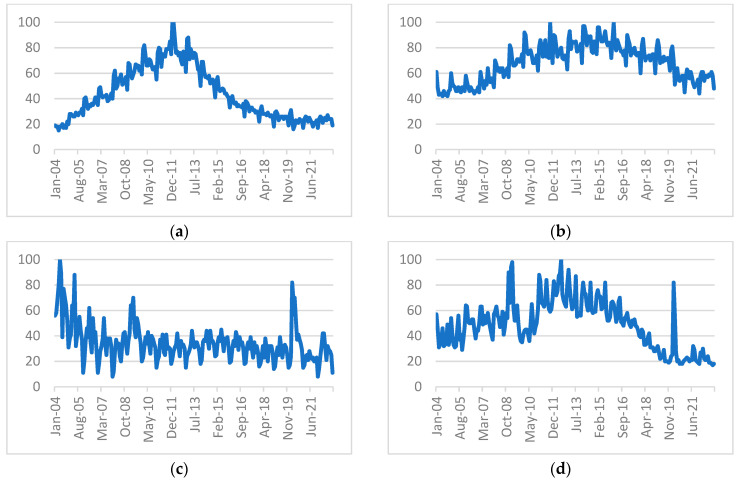
Popularity of certain Google indicators over time: (**a**) Seek (**b**) jobs (**c**) unemployment (**d**) Centerlink.

**Figure 4 entropy-25-01144-f004:**
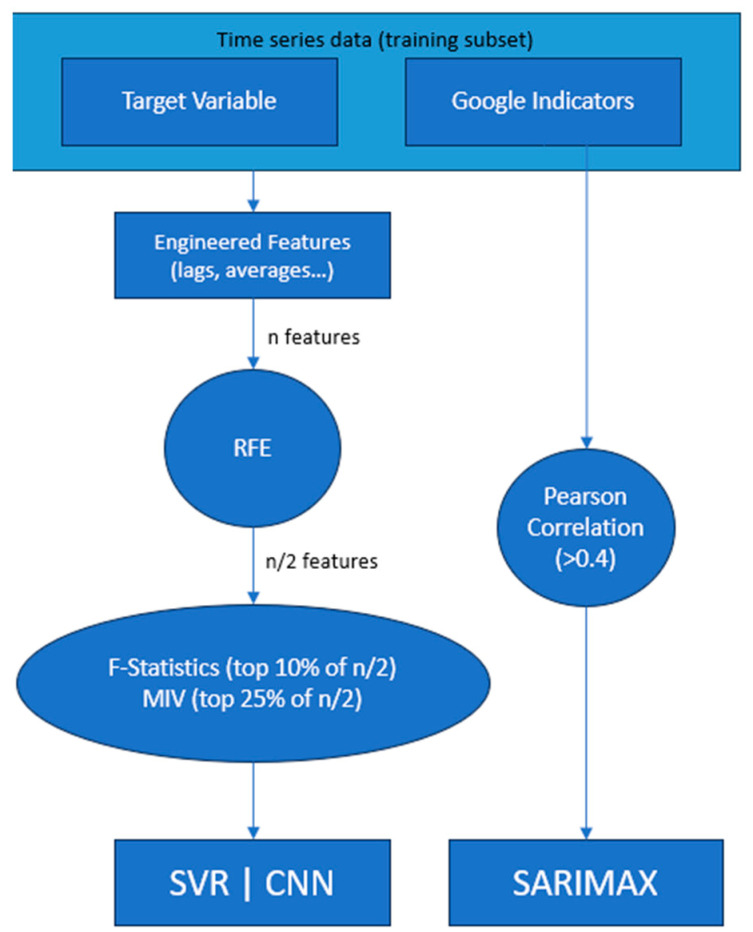
Feature selection approach.

**Table 1 entropy-25-01144-t001:** Selected Google indicators for unemployment rate.

Google Keyword	Type	Reference
Seek	Website	G_U_1
Jobs	Topic	G_U_2
Unemployment	Keyword	G_U_3
Centerlink	Keyword	G_U_4
Indeed	Website	G_U_5

**Table 2 entropy-25-01144-t002:** Selected Google indicators for short-term visitors.

Google Keyword	Field	Reference
Australian Food	Eating	G_U_1
Australia Hotels	Lodging	G_U_2
Australian Map	Traffic	G_U_3
Australian Airline	Traffic	G_U_4
Restaurants in Australia	Eating	G_U_5
Qantas	Traffic	G_U_6
Australia Tourism	Tourism	G_U_7
Places in Australia	Entertainment	G_U_8
Shopping in Australia	Shopping	G_U_9
Australia Weather	Clothing	G_U_10
Climate of Australia	Clothing	G_U_11

**Table 3 entropy-25-01144-t003:** List of time-based features created.

Variable	Features	Description
Date	Month	Month of the year {1,…, 12}
Target variable (y)	y_l_1, y_l_2, y_l_3, y_l_4, y_l_5, y_l_6, y_l_7, y_l_8, y_l_9, y_l_10, y_l_11, y_l_12	Lag value for each of the past 12 months
	y_avg_3, y_avg_4, y_avg_5, y_avg_6, y_avg_7, y_avg_8, y_avg_9, y_avg_10, y_avg_11, y_avg_12	Moving average using 10 different window sizes (3 months,…, 12 months)
Google indicators (G_U_n) (n represents the number of the indicator)	G_U_n_l_1, G_U_n_l_2, G_U_n_l_3, G_U_n_l_4, G_U_n_l_5, G_U_n_l_6, G_U_n_l_7, G_U_n_l_8, G_U_n_l_9, G_U_n_l_10, G_U_n_l_11, G_U_n_l_12	Lag value for each of the past 12 months
	G_U_n_avg_3, G_U_n_avg_4, G_U_n_avg_5, G_U_n_avg_6, G_U_n_avg_7, G_U_n_avg_8, G_U_n_avg_9, G_U_n_avg_10, G_U_n_avg_11, G_U_n_avg_12	Moving average using 10 different window sizes (3 months,…, 12 months)

**Table 4 entropy-25-01144-t004:** List of models created for each experiment and forecasting horizon.

Model Name	Description
SARIMA_HIST	SARIMA using historical data
SARIMA_RECENT	SARIMA using data from 2005
CNN_ TARGET_GI _ALL	CNN using all target and Google data features
CNN_TARGET_ALL	CNN using all target data features
CNN_TARGET_FS	CNN using subset of target data features
CNN_TARGET_GI_FS	CNN using subset of target and Google data features
SARIMAX_ALL	SARIMAX using all exogenous variables
SARIMAX_FS	SARIMAX using subset of exogenous variables
SVR_TARGET_ALL	SVR using all target data features
SVR_TARGET_FS	SVR using subset of target data features
SVR_TARGET_GI_ALL	SVR using all target and Google data features
SVR_TARGET_GI_FS	SVR using subset of target and Google data features

**Table 5 entropy-25-01144-t005:** Description of experiments.

Experiment Name	Description	Data Start Date	Data End Date
Experiment 1	Forecasting monthly unemployment rate with data up to December 2022	Feb-78	Dec-22
Experiment 2	Forecasting monthly unemployment rate with data up to December 2019 (pre-COVID)	Feb-78	Dec-19
Experiment 3	Forecasting monthly number of short-term visitors with data up to December 2022	Jan-76	Dec-22
Experiment 4	Forecasting monthly number of short-term visitors with data up to December 2019 (pre-COVID)	Jan-76	Dec-19

**Table 6 entropy-25-01144-t006:** Train–test split for each set of experiments. * Training Data Start Date for historical models is the start date that corresponds to each experiment.

List of Models	Training Data Start Date	Training Data End Date		Testing Data Start Date		Testing Data End Date	Forecasting Horizon
		Exp (1, 3)	Exp (2, 4)	Exp (1, 3)	Exp (2, 4)		
ARIMA_HIST * ARIMA_RECENT CNN_TARGET_GI_ALL CNN_TARGET_ALL CNN_TARGET_FS CNN_TARGET_GI_FS SARIMAX_ALL SARIMAX_FS SVR_TARGET_ALL SVR_TARGET_FS SVR_TARGET_GI_ALL SVR_TARGET_GI_FS	Jan-15	Sep-22	Sep-19	Data End Date	3	Data End Date	3
	Jan-15	Jun-22	Jun-19	Data End Date	6	Data End Date	6
	Jan-15	Dec-21	Dec-18	Data End Date	12	Data End Date	12
	Jan-15	Dec-20	Dec-17	Data End Date	24	Data End Date	24

**Table 7 entropy-25-01144-t007:** Results for all experiments (MSE).

	Experiment 1				Experiment 2				Experiment 3				Experiment 4			
Model Name	3	6	12	24	3	6	12	24	3	6	12	24	3	6	12	24
ARIMA_HIST	0.0034	0.027	0.0212	3.1912	0.0149	0.0073	0.2801	0.0825	0.6928	1.1835	33.5458	53.2986	0.0002	0.0016	0.0014	0.0023
ARIMA_RECENT	0.0014	0.0031	0.1833	4.7602	0.0146	0.0048	0.4257	0.0836	2.2087	3.0825	1.4924	95.7477	0.0009	0.0017	0.0016	0.0023
CNN_TARGET_GI_ALL	0.0073	0.1903	2.1982	4.1072	0.1432	0.2635	0.1365	0.0837	0.0827	3.6257	7.7866	345.6235	0.0006	0.3542	0.0932	0.1382
CNN_TARGET_ALL	0.0263	0.1666	1.5089	7.9283	0.0019	0.005	0.0069	0.0096	1.1597	1.4089	7.8712	1788.196	0.2724	0.0518	0.0965	0.457
CNN_TARGET_FS	0.1344	0.084	0.3934	2.4659	0.0078	0.0032	0.0034	0.0048	2.384	1.3107	5.5859	1128.699	1.2163	0.0939	0.3582	0.444
CNN_TARGET_GI_FS	0.003	0.1804	0.7338	1.3801	0.0122	0.0469	0.0573	0.1872	0.0259	0.1456	3.8597	464.3315	0.0648	0.4784	1.4217	0.4499
SARIMAX_ALL	0.0003	0.0023	0.087	4.4253	0.0189	0.0122	0.1596	0.0335	1.387	2.7563	0.6647	81.4153	0.0008	0.0006	0.0012	0.0052
SARIMAX_FS	0.0011	0.0032	0.1497	4.6223	0.0142	0.011	0.1581	0.0288	1.6856	2.8412	0.7645	82.3754	0.0008	0.0006	0.0014	0.0045
SVR_TARGET_ALL	0.2168	0.8036	0.762	0.8087	0.0063	0.0054	0.0056	0.0056	1.6918	4.054	8.0787	4.8066	0.0058	0.0051	0.0044	0.0155
SVR_TARGET_FS	0.1334	0.7821	1.1249	0.9628	0.0041	0.0032	0.004	0.0042	1.4593	3.8528	5.9667	2.9439	0.0067	0.0059	0.0058	0.0138
SVR_TARGET_GI_ALL	1.7855	1.8875	1.6613	1.9685	0.0097	0.0487	0.0219	0.0772	0.1438	0.1973	1.1375	3.9978	0.0065	0.0064	0.0055	0.0144
SVR_TARGET_GI_FS	1.6757	1.9403	2.5301	3.4928	0.007	0.004	0.0678	0.1409	0.1386	0.128	0.7697	3.6555	0.0478	0.03	0.0246	0.033

**Table 8 entropy-25-01144-t008:** Feature-selection results.

	Features Selected	Selected Exogenous Variables
Experiment 1	y_l_1; y_l_2; y_avg3; y_avg5; y_l_3; y_l_5; G_U_5_avg12; G_U_2_avg11; G_U_5_avg11; G_U_2_avg9; G_U_4_avg11; G_U_5_avg10; G_U_5_avg8; G_U_5_avg6; G_U_2_avg5; G_U_4_avg9; y_l_10	G_U_2
Experiment 2	y_l_1; y_avg4; y_avg5; G_U_2_avg12; G_U_2_avg11; y_l_5; G_U_2_avg10; y_l_6; G_U_2_avg9; y_l_7; G_U_1_avg7; G_U_2_avg7; G_U_2_avg6; G_U_2_avg8; G_U_5_l_4; G_U_5_l_5; G_U_5_avg3	G_U_2; G_U_5
Experiment 3	y_l_12; G_U_4_avg4; G_U_6_l_2; y_l_4; G_U_2_l_3; G_U_7_avg3; G_U_6_l_8; G_U_6_l_5; G_U_4_avg10; G_U_4_avg6; G_U_6_avg4; G_U_6_l_7; G_U_6_avg9; G_U_4_avg5; G_U_3_avg4; y_l_1; G_U_6_avg12; G_U_7_avg4; G_U_7_l_3; G_U_2_l_8; G_U_7_l_2; G_U_4_l_4; G_U_3_l_3; G_U_3_avg6; G_U_4_avg8; G_U_2_l_4; G_U_4_avg12; G_U_3_avg7; G_U_3_avg9; G_U_4_l_3; G_U_2_avg8; G_U_2_l_2; G_U_3_l_5; G_U_5_avg3; G_U_9_avg3; G_U_9_avg4; y_l_7; G_U_9_avg5; G_U_5_avg5; G_U_6_avg5	G_U_5; G_U_6; G_U_8; G_U_9
Experiment 4	y_l_12; y_avg4; G_U_10_l_4; G_U_6_l_3; y_l_4; G_U_6_l_2; G_U_10_avg8; y_avg3; G_U_6_avg3; G_U_10_avg9; G_U_4_avg4; G_U_7_avg3; G_U_3_l_5; y_avg6; G_U_4_l_8; G_U_10_l_5; G_U_4_l_4; y_l_9; G_U_7_l_2; G_U_4_l_2; G_U_6_l_5; G_U_4_avg5; G_U_6_l_8; G_U_2_l_4;	G_U_1; G_U_2; G_U_3; G_U_4; G_U_6; G_U_7; G_U_8; G_U_9; G_U_11

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
