# Peer review of "A Model Selection Approach for Time Series Forecasting: Incorporating Google Trends Data in Australian Macro Indicators"

_entropy, 2023, doi:10.3390/e25081144_

Round 1

Reviewer 1 Report (Previous Reviewer 1)

Dear authors,

That work studies the impact that Google Trend indicators may have in time series forecasting of two representative use cases. Although the results seem promising, despite the existing inconsistencies between the performance of linear and non-linear models for different time horizons, there are some critical issues that do not let me suggest that manuscript for publication.

- No code / implementation is provided, thus my confidence over the presented results is low.

- The examined models do not cover important families of algorithms that have recorded great performance in similar tasks (e.g., ensemble models, deep-learning variants, prophet model).

- Most of the applied stages (feature selection / dimensionality reduction / feature engineering) are not validated.

- The outcome of the feature selection process should be further discussed. Moreover, your dataset is not revealed or summarized so as to understand better the raw data. 

- Additional performance metrics are needed. 

Based on those comments, both the experimental framework and the previous Sections should be clearly enriched.

Well-documented manuscript.

Author Response

Reviewer 2 Report (New Reviewer)

Many details of the study are obscure and turn it difficult to reproduce. For instance, the software employed isn't mentioned.

The manuscript needs careful proofreading

Round 2

Reviewer 1 Report (Previous Reviewer 1)

Dear authors,

Thank you for your answers. However, all the discussed critical concerns from my side are pending:

- I do not see the code, thus I cannot trust your results.

- By writing that your stages need validation, I mean that we have to see comparisons with the competitors of each stage (e.g., PCA vs uMAP)

- Since one reviewer asks to add another one metric, you should follow his/her advice. 

- It is difficult for the reader to reproduce your method, which is not an asset for the research community.

Round 3

Reviewer 1 Report (Previous Reviewer 1)

Dear authors,

Thank you for attaching your code after the previous revision. Now we can start discussing over the confidence and the quality of that draft.

First of all, you have to create a public repo with structured code and information about the scope and the operation of your code (e.g., GitHub repo). Empty cells and lists with tens of manually added items must be removed, as well as include requirements.txt for running your experiments.

Secondly, you have to introduce statistical comparison of your methods, as well as some visualizations as it regards the performance of the models.

Next, I see your opinion about not comparing tens of methods that can be found in the literature for pre-processing, but I did not ask to make extensive comparisons. However, checking through grid search the best combination of features is not a sophisticated approach, and should be replaced with better frameworks (e.g., fuzzy feature selection). Moreover, selecting for example just the 3 first components of PCA without any validation does not help the community to understand the effectiveness of that method through a fixed choice.

Finally, the correlation of the input variables with the target variable is needed to be provided, so as to understand better the underlying relatiionship of the data.

Clear and concise.

Author Response

Thank you for the review. We have revised the paper to address your comments. 

Round 4

Reviewer 1 Report (Previous Reviewer 1)

Dear authors,

Your work has been improved based on the last revision round. However, there are still some critical points that should have been resolved after 3 rounds of revisions.

- 'The repository includes well-structured code with a comprehensive README file, and a requirements.txt file for ease of replicating our experiments. All empty cells and redundant items have been removed, and we will continue to update and improve this repository' -> This statement does not hold. Your repo is far away from what you have mentioned. You have to update the readme, create a requirements.txt file (e.g., python - Automatically create requirements.txt - Stack Overflow). Comments should be added to your code, as well as elaborate the structure of your repo (different folder for data, and split the notebooks as you did in the provided files). These steps are too simple to have left them unchanged after my previous comments.

-  'Due to the extensive amount of results data, it wasn't feasible to include all the results directly in the paper' -> Statistical comparison is necessary to be added and discussed into the main body of the paper. References and tables have also to be added so as to be consistent. There are specific plots that can facilitate you. Otherwise mention the statistically non significant pairs that you detected. The code also of the statistical analysis is necessary or the instructions for computing that in any platform needed.

- 'Feature Selection: In response to your feedback, we replaced PCA with Recursive Feature Elimination (RFE) using a Decision Tree Estimator for feature selection' -> I did not ask the replace the previous approach, but just to justify the selection of specific steps / values. I suggest retrieving the previous approach and compare it with the existing which is more dynamic. Any conclusion based on results is useful.

- Section 2: Literatures -> Related Work

- The quality of Figure 1 and Figure 2 needs to be fixed. X-axis ticks need to be rotated. X-axis ticks on Figure 3 need revision.

- Describe the role of Reference column in Table 1, and ideally change the name of the column, or even replace G_U_n with GUn. Generalize that changes across the manuscript.

- 'Creating a training dataset' is not a proper name for the first step of a proposed algorithm.

-Table 7 is poor, regarding both its format and its readability. Highlight the best value per experiment and depict the average performance of each algorithm through boxplots or any other descriptive plot. Moreover, depict the same table for the other performance metric.

- The use of Prophet seems more appropriate against the proposed method, since this is the state-of-the-art model for such cases. If you have the appropriate time, include this in your comparisons.

In any case, you have to be really careful revising all of the above points.

Adequate use of English Language.

This manuscript is a resubmission of an earlier submission. The following is a list of the peer review reports and author responses from that submission.

Round 1

Reviewer 1 Report

Dear authors,

That work investigates the utilization of data and metrics retrieved from internet APIs along with historical data, and applies them under different configurations to linear and non-linear regression models. The research is limited to Australian case, reviewing at the same time other similar approaches, conducted mainly on a national level. Two different indicators are used as target variables, while some end-to-end pipelines are formatted for measuring the effectiveness of those methods under two different performance metrics. Although this idea appears interesting, there are important flaws and issues. I mention some of them:

- You repeat the same terms and phrases several times, even into the Abstract. 

- The first two sections need to be rephrased since they suffer from the above issue.

- Various definitions are well-known and almost trivial (AR, MA, SARIMA) and could be avoided, providing more context on the implemented parts by your side.

- The literature review part should be summarized so as not to read continuously the same phrase, but for highlighting possible discrepancies in the applied methods and common lessons learned from previous work / surveys. Some Tables could be introduced, presenting those findings more effectively. That part was almost boring, and did not mention more dedicated information.

-  You do not describe explicitly the indicators / features / variables retrieved from Google Trends and used into your models. The experiment should be reproduced by anyone who reads your manuscript. This means that you should provide more detailed analysis, as well as providing the data (if this is feasible) and your implementation.

- The pre-processing stage seems shallow. Didn't you apply any feature transformation other than log transformation? What about the month variable, did you leave it untouched? More details are needed there.

- The selection of the regression models is poor. You should include various ML-based models and other SOTA approaches (Prophet, LSTMs, Wavelet-based models).

- The presentation of the results should be accompanied by statistical importance and some visualizations, apart from the useful ranking at the end of the manuscript.

- The selected features per case should be discussed and recorded, probably with histogram or other data visualization tools.

Reviewer 2 Report

Advantages.

The topic of this article is interesting and meaningful for specific Australian problems. This is a good analytical report with practical experiments implementation. 

 0.The abstract elements of the article are incomplete and need to be modified according to the following: purpose - research methods - research findings - research conclusions.

Keywords must be in the alphabetical order.

 1.Introduction.

Acceptable.

 2.Literature.

In my opinion, literature review is excellent.

 3.Techniques

The mathematical part is acceptable.

 4. Experimental data

Methodology for experimental part is given (extract indicators and so on)

 5. Approach

Two models are described.

(found small mistake – duplicate figure 4)

 6. Results

Approx. 4 pages of real experimental research.

Very, very old data for experimental part (until 2016)

 7.Conclusion

Conclusion part is acceptable.

Disadvantages:

Parts 1,2,3,4,5 are well structured and is topically.

BUT! The results of the experimental part are no longer relevant because old data is used.